# Non-Rhizobial Endophytes (NREs) of the Nodule Microbiome Have Synergistic Roles in Beneficial Tripartite Plant–Microbe Interactions

**DOI:** 10.3390/microorganisms13030518

**Published:** 2025-02-26

**Authors:** Ahmed Idris Hassen, Esther K. Muema, Mamonokane O. Diale, Tiisetso Mpai, Francina L. Bopape

**Affiliations:** 1ARC-Plant Health and Protection, P. Bag X134, Queenswood, Pretoria 0121, South Africa; dialemo@arc.agric.za (M.O.D.); mpait@arc.agric.za (T.M.); phalanef@arc.agric.za (F.L.B.); 2Department of Plant and Soil Sciences, Faculty of Science, Engineering and Agriculture, University of Venda, Thohoyandou 0950, Limpopo, South Africa; 3Department of Soil Science, Faculty of Agri-Sciences, Stellenbosch University, Stellenbosch 6201, Western Cape, South Africa; emuema@sun.ac.za

**Keywords:** rhizobia, nitrogen fixation, endophytes, nodules, microbiome, metagenomics, synergism

## Abstract

Microbial symbioses deal with the symbiotic interactions between a given microorganism and another host. The most widely known and investigated microbial symbiosis is the association between leguminous plants and nitrogen-fixing rhizobia. It is one of the best-studied plant–microbe interactions that occur in the soil rhizosphere and one of the oldest plant–microbe interactions extensively studied for the past several decades globally. Until recently, it used to be a common understanding among scientists in the field of rhizobia and microbial ecology that the root nodules of thousands of leguminous species only contain nitrogen-fixing symbiotic rhizobia. With the advancement of molecular microbiology and the coming into being of state-of-the-art biotechnology innovations, including next-generation sequencing, it has now been revealed that rhizobia living in the root nodules of legumes are not alone. Microbiome studies such as metagenomics of the root nodule microbial community showed that, in addition to symbiotic rhizobia, other bacteria referred to as non-rhizobial endophytes (NREs) exist in the nodules. This review provides an insight into the occurrence of non-rhizobial endophytes in the root nodules of several legume species and the beneficial roles of the tripartite interactions between the legumes, the rhizobia and the non-rhizobial endophytes (NREs).

## 1. Introduction

The world’s population has been projected to increase from 8.1 billion in 2024 to around 9.7 billion by 2050. Coupled with this, the increasing threats of global climate change effects and other environmental impacts, such as soil abiotic stresses and drought, are among the greatest challenges agriculture faces in terms of achieving the goal of global food security. It is known that the Green Revolution, following the pioneering invention of the Haber–Bosch process at the beginning of the 20th century, provided a solution for improved agricultural nutrients with the production of inorganic fertilizers to feed the world [1]. However, the prolonged usage of inorganic fertilizers and chemicals has caused huge negative environmental impacts globally, including a decline in biodiversity, increased greenhouse gas emissions, and deteriorating conditions in terms of soil health [2]. This demands a sustainable means of farming practices that maintain soil health, increase agricultural productivity and sufficiently feed the world. One of the major alternative means of sustainable intensive agricultural practices currently being emphasized to solve the problems associated with soil health and global food security is the use of beneficial soil microbes as alternatives to inorganic fertilizers and chemicals. Among such beneficial soil microbes are the rhizobia, which interact and form symbiotic associations with several hundreds of plant species within the Leguminosae family and are capable of fixing atmospheric nitrogen (N_2_) by forming nodules. Nodules are specialized organs formed on the roots or stems of legumes through a symbiotic relationship with soil rhizobia [3,4]. Nitrogen-fixing symbiosis begins with a unique molecular dialogue between the plants and the rhizobia following the release of certain polyphenolic compounds called flavonoids by the legume roots into the rhizosphere, a signal communication triggered by nitrogen nutrient limitation in soils. As soon as the rhizobia perceive the flavonoids, they respond by activating the transcription of symbiosis-related (Nod) genes, which is orchestrated by the activity of the nodulation protein D (NodD). The expression of the nod genes results in the secretion of lipo-chitooligosaccharide (LCO) compounds called the Nod factors, signal molecules essential for initiating symbiotic development, root hair curling, induction of nodulation and entry of the rhizobia into the nodules [5]. It is inside the nodules that endophytic rhizobia transform atmospheric nitrogen (N_2_) into forms that are accessible to plants [6,7]. In return, the legumes supply their rhizobial microsymbionts with photosynthetic products in the form of nutrients.

There are still contrasting reports on whether the symbiotic nitrogen-fixing rhizobia are the dominant population in the nodules as compared to the non-rhizobial endophytic bacteria. Although there are reports that rhizobial symbionts dominate the interior of the nodules and outcompete other bacteria to efficiently communicate with the host during infection, refs. [8,9] detected about 99% and up to 87% of the total bacterial population to belong to non-rhizobial endophytes, respectively. However, as several culture-independent studies pooled multiple root nodules per plant or nodules from different plants, it raised a question as to whether the observed diversity and relative abundance of NREs are uniformly distributed across all the root nodules [10].

Although traditional culture-dependent methods can be used to identify the microbiome inside the root nodule, the diversity identified will be influenced by various factors, which include the selection of medium, incubation and bacterial growth conditions. In addition, ref. [11] stated that more than 95% of environmental microbial communities cannot be isolated by culture-based techniques. On the contrary, culture-independent techniques such as the metagenomics approach have been used to shed light on the microbial community of the interior of the nodules by allowing the detection of both rhizobia and non-rhizobial endophytic bacteria in the nodules [9]. This technique helps in the characterization of unculturable microbes, provides empirical data on the entire microbial diversity, and provides full genetic information on the microbes and their relative abundance [12,13]. It also provides a novel and important approach to identify the rare or less abundant microbial communities. Several metagenomics studies have reported on NREs isolated from the root nodules of different species of legumes that harbor a diversity of microbiomes, such as *Pseudomonas*, *Niastella*, *Shewanella*, *Ohtaekwangia* and *Rhizobacter* species, alongside the nitrogen-fixing bacteria *Ensifer* [14,15]. Even though the NREs do not induce nodules, they still offer advantageous traits to their host plant, such as plant growth promotion, environmental stress resistance, disease protection, as well as nodulation enhancement [16,17]. A question also worth raising is how these non-rhizobial endophytes gain access to and manage to enter the interior of the nodules. In this review, we highlight the occurrence of non-rhizobial endophytes in the root nodules of the once thought to be highly host-specific legumes and the beneficial roles they play in legume–rhizobium symbiosis.

### Review Methodology

This review was prepared by articulating the research question first and performing preliminary research through reading and analyzing various literature works that helped the authors with a broader understanding of what non-rhizobia endophytes constitute and their wide occurrence in the nodules of several leguminous plants. We conducted a comprehensive search of the relevant literature related to this review using search engines that included Web of Science, Science Direct and BIOSIS Previews. By employing major keywords such as non-rhizobia endophytes, nodules, microbiome, nitrogen fixation, plant growth promoting rhizobacteria, and metagenomics, a total of 150 research articles were initially searched and downloaded. Having excluded all the redundant and less important articles, the final 109 articles were used as sources of information in preparing this review.

## 2. Signaling in Rhizosphere Plant–Microbe Interactions

### 2.1. Microbe–Microbe and Soil–Microbe Signaling

Microbial communities in the rhizosphere are dominated by their own microbial interactions, among which the most complex interactions come from the root-associated microorganisms in the rhizosphere, as compared to that of the bulk soil. The commonly known microbe–microbe interaction in the rhizosphere occurs by way of molecular signaling known as quorum sensing (QS), a cell-to-cell signaling in bacteria important in group-coordinated processes [18]. Among the diversity of quorum-sensing signals produced by many soil microbes are the N-acyl homoserine lactones (AHLs), diffusible signal factors (DSFs) and diketopiperazines (DKPs) [18,19]. Soil bacteria use quorum sensing for rhizosphere colonization, rhizosphere competence, virulence, and production of secondary metabolites [20]. One of these QS signals can also function as an inter-kingdom signal to elicit various effects on plant development and plant immune responses, such as induced systemic resistance (IRS).

### 2.2. Plant–Microbe Signaling

It is indicated that direct root microbe interactions are essential for plant growth and soil health [20]. The best-deciphered and extensively studied plant–microbe signaling network in the plant rhizosphere is the interaction that occurs in legume–rhizobium symbiosis. The legume plants produce diffusible molecules called flavonoids that induce the bacterial nod genes [21,22]. These nod genes produce and secrete lipo-chitooligosaccharides (LCOs), also called nodulation (Nod) factors. The LCOs are then perceived by the roots through receptor kinases at the root epidermis and activate signal cascades, leading to the formation of nodules [22,23]. Apart from this signaling in the legume–rhizobium association, other rhizosphere microbes are also capable of influencing their plant host by releasing diverse signaling molecules (Figure 1).

An essential gateway for plants to take up water and nutrients from the soil is the root–soil interface. Plant roots exert their effects on soil through rhizodeposits. In these root–soil interactions, however, a multitude of microbes actively participate, such as arbuscular mycorrhizal fungi (AMF), which colonize about 80% of terrestrial plants and help plant roots to uptake soil phosphorous. Other rhizosphere bacteria can fix nitrogen, solubilize and release plant available phosphorous, potassium and other micronutrients to be taken up by the roots [24]. The root exudate is the major factor that chemically and physically differentiates between the rhizosphere soil and the bulk soil. When rhizosphere microbes assimilate rhizodeposition, the soil quality is highly improved. A good example is described in [25], in which the assimilation of root polysaccharides by *Bacillus subtilis* induced the production of a biofilm matrix that enhanced root colonization and plant beneficial effects, maintained the soil moisture and facilitated soil aggregation.

## 3. How Do NREs Enter the Nodules?

Non-rhizobial endophytes (NREs) are bacteria, other than the symbiotic nitrogen-fixing rhizobia, which actively colonize the root and occur inside the nodules of leguminous plants along with the rhizobia. Although the legume–rhizobium interaction has long been known to be highly specific due to the selective nodule environment, several species of non-rhizobial nodule endophytes have been reported. However, such findings that non-rhizobial bacteria were isolated from within the nitrogen-fixing nodules were not readily accepted at first [26]. Later, after the initial report of the isolation from the nodules of a nodulating β-proteobacterial strain in 2021, the number of reports published on the occurrence of such bacteria within the nodules of legumes increased exponentially [26]. It was then disclosed elsewhere that the once traditionally exclusive niche of rhizobia, the root nodule, is being colonized by non-rhizobial microbes not related to symbiotic nitrogen fixation. For instance, as many as 4.3 × 10^9^ cfu/g of rhizobia and 3 × 10^5^ cfu/g non-rhizobial endophytes belonging to 12 genera were detected per gram of fresh weight of red clover nodule tissues [27]. The key question is as follows: how do these rhizobacteria breach the legume host specificity and enter the root nodules? There are several hypotheses to explain how non-rhizobial endophytes colonize the roots and make their way to the interior of the nodules.

According to some reports, NREs randomly penetrate the nodule together with their rhizobia counterparts, hence not being preferentially selected by host plants [10]. Moreover, they occur in lower abundances than the preferred rhizobia microsymbiont of host plants and due to this, it is suggested that their roles in plant growth promotion may be limited. However, according to a more experimentally proven hypothesis, the host-specific rhizobium initiates the signaling process and forms infection threads (ITs). Rhizobacteria in the vicinity invade the ITs and enter the nodules (Figure 2). This hypothesis is supported by studies that involve advanced fluorescent microscopy coupled with a visual approach with molecular tagging of rhizobacteria [28,29]. Some studies indicate that the invasion of infection threads and the entry into the nodules by non-rhizobia endophytes is not a random process. Plants have evolved a sophisticated surveillance system for monitoring microbial invasion and the corresponding response strategies, resulting in only a limited range of non-rhizobial endophytes being able to enter the plant tissues [30,31].

Many researchers believe that although the NREs are not capable of inducing nodule formation, they are able to enter infection threads when co-occurring with nodule-forming rhizobia [32]. An earlier study reported an aquatic NRE associated with *Neptunia natans* to be able to enter the primary root cortex through natural wounds caused by the splitting of the epidermis as well as the emergence of young roots, and thus, to stimulate the root nodules [33]. According to the report by [33,34], NRE diversity in root nodules is more influenced by the native soil microbial community rather than the plant genotype. However, ref. [35] argued that NRE access into and inhabitancy of the legume root nodule are strictly and selectively regulated by the legume host rather than rhizobia signaling.

These researchers indicated that the colonization of *Lotus japonicus* nodules by the endophytic bacteria *Herbaspirillum* sp. B501, *Rhizobium mesosinicum* KAW12, and *Burkholderia* sp. KAW25 involves one of the following three major processes: (i) non-rhizobial endophytes’ nodule occupancy is host-controlled since the initial infection threads (ITs) are initiated by the legume host, (ii) exopolysaccharides represent key bacterial features for the chronic infection of nodules, or (iii) the plant’s symbiotic genes, such as *Cyclopes*, *Cerberus*, *Nep1*, and *ArpC1*, are involved in signaling pathway and control the invasion of nodules by both symbiotic and endophytic microbes.

## 4. Metagenomics and Molecular Studies of the Nodule Microbiome

Several studies of the nodule microbiome have made use of the shotgun or 16S metagenomics sequencing protocol to determine the diversity of symbiotic and non-rhizobia endophytes within the nodule. In a study by [36] on a nodule microbiome of cowpea and lima bean using metagenomics, although the nodule microbiome was predominantly occupied by *Bradyrhizobium* spp. (90%), there was wide bacterial diversity, including the group *Bacillales*, *Actinomycetales*, *Firmicutes*, and several others belonging to the Alpha and Beta Proteobacteria and Gama Proteobacteria, to mention a few. Like many other legumes, Actinorhizal plants also form a symbiotic association, but with other groups of nitrogen-fixating soil bacteria, the *Frankia*, which form nodules in roots. There are also several reports on the occurrence of endophytic bacteria in the root nodules of these plants. Just like the nodule microbiomes in legumes, the microbial composition, richness, and diversity of the root nodules were reported to vary significantly across 18 Actinorhizal plants, as studied by shotgun metagenomics [37]. The analysis revealed that the nodules were not only harbored by the *Frankia* but also contained endophytic bacteria belonging to *Enterobacter*, *Pseudomonas*, *Streptomyces* and many other species (Figure 3).

In a case study with *Acacia longifiola*, one of the most aggressive legume invaders that forms symbiosis with rhizobia, a nodule microbiome sequence analysis revealed the presence of several genera of microbes, including cyanobacteria and fungi [38]. In another study of the nodule microbiome of *Medicago polymorpha* by [39], the presence of broader microbial populations within the nodules was detected. Analysis of the 16S rRNA gene sequences extracted from the shotgun metagenomics showed that, in addition to the dominant genus *Ensifer*, non-rhizobial endophytic species belonging to Firmicutes and Actinobacteria were detected. Non-rhizobial endophytic strains such as *Enterobacter*, *Pseudomonas* and *Stenotrophomonas* species were detected based on 16S rRNA sequence analysis of the nodule extracts in cowpea (*Vigna unguiculata*) [40]. A similar study by [41] reported the occurrence of *Bacillus endophyticus*, *Kocuria* sp. and *Brevundiminas* spp. in the root nodules of soya bean; *Streptomyces* sp., *Bacillus* sp., and *Actinobacteria* sp. in the nodules of mung bean; and *Bacillus endophyticus* in cowpea nodules. The majority of these non-rhizobial endophytes also showed positive ecological and plant growth-promoting traits, as well as phosphate solubilization activities. In a study that involved faba bean (*Vicia faba* L.) by [42], 34 non-rhizobial endophytes that failed to nodulate their original host were isolated from the nodules. These nodule endophytes were identified as members of the Enterobacteriaceae. Although several members of this group are associated with human and animal intestines as pathogens, certain Enterobacteriaceae, such as *Enterobacter* and *Citrobacter*, have been reported to form beneficial relationships with plants [43].

## 5. Role of NREs in Plant Growth and Mitigation of Abiotic Stress

Abiotic stresses such as drought, extremes of temperatures, salinity, acidity and metal toxicity not only limit the growth of several agricultural crops worldwide but also contribute a lot to the significant decline in soil health. Many non-rhizobial endophytes detected in the nodules of several legume species have at least one beneficial trait that helps plants tolerate conditions of abiotic stress. In the following sections, some of the most commonly known plant growth promotion and abiotic stress-tolerant traits of NREs are discussed.

### 5.1. ACC Deaminase Activity

Although ethylene is a significant plant hormone required at some stage in the growth of plants, such as breaking seed dormancy and root development, a high level of ethylene has some inhibitory effect on root elongation following germination. While the level of ethylene varies in plants depending on the environmental conditions, it is active at a low level of 0.05 μg L^−1^ under stress-free conditions. However, under severe stress conditions, the ethylene level reaches a dangerous level of 25 g L^−1^ and results in deleterious responses such as growth repressive effects, including senescence, retarded seed germination and root development, leaf abscission and loss of chlorophyll pigment [44,45]. Several plant growth-promoting rhizobacteria (PGPR) and many species of rhizobia produce the enzyme ACC (1-amino-cyclopropane-1-carboxylate) deaminase, which decreases the level of ethylene associated with stress in plants by cleaving the ethylene precursor ACC into ammonia and α-ketobutyrate. ACC deaminase, encoded by the *acdS* genes, may not be found in all the strains within a particular species but are detected in many rhizosphere bacteria and rhizobial species, including α- and β-rhizobia [46].

The inhibitory effect of ethylene in the nodulation process of the legume–rhizobium symbiosis and the subsequent reduction in nitrogen fixation are well documented [47,48]. Moreover, ref. [49] reported the negative role of ethylene in the nodulation process that occurs during the legume–rhizobium interactions by inhibiting the formation and functioning of nodules. Salinity is one of the major abiotic stresses that results in the impaired symbiotic performance of several legume species by inhibiting seed germination, seedling growth, vigor, and flowering due to the accumulation of stress ethylene. ACC-deaminase- positive PGPR reduces the level of stress ethylene and confers tolerance under salinity stress [50]. ACC-deaminase-positive strains of *Paenibacillus* sp. ACC-06 in *Phaseolus vulgaris* [51], *Bacillus*, *Acinetobacter* and *Enterobacter* spp. in *Medicago sativa* [52], *Variovorax paradoxus* 5C-2 and *Rhizobium leguminosarum* in *Pisum sativum* [53] have all mitigated the negative impact of salinity stress and promoted effective nodulation and plant growth under salinity stress conditions.

In addition to their symbiotic associations with rhizobia, legumes also establish associations with non-rhizobial endophytes, many of which can synthesize the enzyme ACC deaminase. Several studies have demonstrated the importance of ACC deaminase for the symbiotic rhizobial–legume interaction, particularly under different abiotic stresses [54]. According to these researchers, exogenous ACC deaminase production by the endophytic *Pseudomonas* sp. Q1 promoted the symbiotic performance in the pasture legumes *Trifolium subterrraneum* and *Medicago polymorpha* when co-inoculated with the microsymbionts *Rhizobium leguminosarum* and *Ensifer meliloti* under a high manganese concentration. In another study, an ACC-deaminase-positive endophytic *Serratia grimesii* BXF1 was able to promote early nodulation and growth in *Phaseolus vulgaris* [55]. The existence of a synergistic relationship between an ACC-deaminase-producing *Pseudomonas putida* strain PSE3 and *Rhizobium leguminosarum* was reported by [56], in which dual inoculation of *P. putida* and the rhizobium resulted in increased symbiotic efficiency and growth of *Pisum sativum* in both pots and field experiments.

Some studies have elucidated the modes of actions of how salinity stress is mitigated by the ACC deaminase activity of microbes. For instance, ACC-deaminase-producing bacteria can confer salt resistance by increasing membrane transport systems such as the Na+/H+ pump and several other ionic channels [57]. Moreover, bacterial-induced changes in the transcription factor (TF) machinery are known to modulate the expression of stress-related genes [57,58]. The process of nodulation and biological nitrogen fixation in legumes is not only affected by drought and salinity stresses. Heavy metal stress and waterlogging are also known to cause deleterious effects on plant growth and development. During flooding, many plants, including legumes, experience anoxia (lack of oxygen) that triggers higher production of the ethylene precursor ACC and hence of ethylene. Several studies showed that the nodulation capacity of hypoxia-sensitive legumes decreased with reduced nodule weight when grown under hypoxic conditions. Hypoxia-sensitive legumes, such alfalfa [59] and soybean [60], exhibit reduced nodule weight when grown under hypoxic conditions. *Medicago truncatula* nodulation shows a 45% decrease under 0.1% O_2_ but is not affected by 4.5% O_2_ treatment, and the nodule fresh weight per plant is not dampened by 4 weeks of hypoxia [61].

The inoculation of ACC-deaminase-producing *Streptomyces* sp. GMKU 336 in rice seedlings resulted in reduced levels of ethylene and improved root elongation and biomass production under waterlogging stress [62]. Production of stress ethylene induced by high concentrations of heavy metals limits root growth and development in many plant roots. One of the microbial solutions to mitigate this effect includes the utilization of metal-tolerant ACC-deaminase-producing strains of PGPR known to optimize plant growth in heavy-metal-stressed soils [63,64]. In summary, apart from the mitigation of different types of abiotic stresses in various species of plants, ACC deaminase activity is essential in the synergetic interaction between rhizobia and the endophyte, as it positively contributes to an efficient legume–rhizobia symbiosis by regulating the inhibitory ethylene levels that inhibit the process of nodulation and retard the overall plant growth.

### 5.2. Other NRE Traits for Mitigation of Abiotic Stress

Many strains of NREs found in the nodule microbiomes have once been part of the active free-living rhizosphere microorganisms with several PGPR properties (Figure 4). For instance, metagenomic sequence analysis of the soybean nodule microbiome and the rhizosphere soil revealed that almost all the prokaryotic sequences detected in the root nodules were also found in 276,338 sequences from nine rhizosphere soil samples [10]. This could be because, under certain favorable conditions, the non-symbiotic, non-nitrogen-fixing and free-living rhizosphere strains could have made their way into soybean root nodules and become non-rhizobia endophytes (NREs). As indicated in Figure 4, there is an interplay between several groups of plant growth-promoting rhizosphere bacteria and the plant roots under various abiotic-stressed conditions. In this interaction, various types of abiotic stresses, such as drought, salinity and fertility stress, could be mitigated by the production of certain bioactive compounds by the PGPR residing on the root surfaces. For instance, the secretion of cytokinin by the bacteria can result in the accumulation of abscisic acid (ABA) in the leaves, which elicits stomatal closure under drought stress. By contrast, the secretion of antioxidants and the enzyme ACC deaminase by PGPR results in the degradation of reactive oxygen species (ROS) and that of the ACC precursor, respectively [65]. These processes elicit induced systemic tolerance (IST) in the plants against drought and salt stress conditions and promote plant growth indirectly.

It means that, in addition to ACC deaminase activity, plant growth-promoting rhizobacteria, most of which also exist as non-rhizobial endophytes, elicit induced systemic tolerance (IST) against drought, salt and other abiotic stresses and promote plant growth indirectly [65]. For instance, co-inoculation of *Bradyrhizobium japonicum* USDA-110 with *Pseudomonas putida* NUU8 in soya beans resulted in IST and increased growth and nodule number under drought stress conditions [66]. It is known that a high concentration of ROS causes oxidative stress in plants, leading to damage in various cellular components. Production of cytokinin and antioxidant molecules by PGPR may result in the accumulation of abscisic acid (ABA), which in turn results in the degradation of reactive oxygen species (ROS). ROS in plants can be triggered by a number of abiotic stresses, including salinity, drought, flooding or heavy metal contamination such as mercury, chromium, arsenic and lead that cause high toxicity in plants [67]. In a study by [68], the negative impact of osmotic and oxidative injuries in soya bean was mitigated by inoculating a strain of *Azotobacter chrococcum* that produces high levels of anthocyanins and other antioxidants. Under high salt concentrations, some PGPRs also produce volatile compounds that downregulate the HTK1 gene expression in roots and lower the level of Na+, thereby reducing salinity-associated stress [65,69]. Various pieces of evidence exist that show the importance of the root microbiome in terms of plant health, and it is evident that plant roots influence and control the composition of the rhizosphere microbiome due to their unique rhizodeposition, which attracts a unique group of microbes. Generally, beneficial plant–microbe interactions, such as the synergistic interaction between free-living rhizosphere microbes and the plant root and the soil, as well as the interaction that exists in legume–rhizobium symbiosis, are the driving force behind the maintenance of soil health and the improvement of plant growth and productivity [70].

## 6. Plant Growth-Promoting Traits of NREs in Legumes

There are a greater number of cells and greater microbial diversity in the root rhizosphere as compared to the rootless soils due to the presence of nutrients exuded by the plant roots, including sugars, amino acids, organic acids and other small molecules representing up to a third of the carbon the plants fix [71]. In return for these rhizodeposition that the microbes use for their growth, they render quite a significant number of beneficial functions to the plants through their versatile enzyme activities and plant growth promoting traits, of which we briefly discuss a few selected few.

### 6.1. Indoleacetic Acid (IAA)

The production of indole acetic acid (IAA) is one of the direct modes of action by which soil microbes promote plant growth. The major effects of IAA produced by microorganisms include enhanced root growth and improved water and nutrient uptake by plants, which can also help plants to cope with drought stress [72,73]. IAA is reported to be produced by various species of NREs belonging to *Pseudomonadaota*, *Actinomycetota*, *Bacillota*, *Bacteroidota* and *Ascomycota* phyla. The NREs that showed IAA production in various legumes were recorded for *Glycine max* [74], *Vicia faba* [75], *Pisum sativum* [75], *Cicer arietinum* [76], *Arachis hypogaea* [77], *Pisum sativum* [78] *Vigna radiata* [79,80] and *lens culinaris* [81]. These reports highlighted that the production of IAA by NREs residing in the nodules of the different legumes improved the plant heights, root lengths, plant biomasses and yields [82].

### 6.2. Phosphorus

As the symbiotic nitrogen fixation that occurs in the legume–rhizobium symbiosis is a high-energy-demanding (ATP) process, the soil available phosphorous (P) is very essential to the growth and yield of legumes. In fact, phosphorous is one of the plants’ limiting nutrients after nitrogen. In addition, it is one of the limiting nutrient elements required by plants as it is involved in several roles, including nitrogen fixation, photosynthesis, energy transfer and signal transductions, to mention a few [83]. Most agricultural soils have large amounts of inorganic and organic phosphates, with 95–99% of insoluble phosphorous present in the soil. However, P always becomes immobilized and unavailable to plants due to its high reactivity with some metal complexes leading to precipitation or adsorption to the soil [84,85]. Even if it is possible to correct ‘P’ deficiency by applying ‘P’ fertilizer, it is quite often unaffordable by most resource-poor farmers in the tropics and subtropics. The use of phosphate-solubilizing microbes as biofertilizers is a promising approach to improve food production and increase crop yields. Phosphate-solubilizing microorganisms can promote plant growth by enhancing the efficiency of biological nitrogen fixation and phytohormone synthesis and increasing the availability of certain micronutrients, such as zinc and iron [86,87]. In light of this, the use of consortia of rhizobia and phosphate-solubilizing bacteria (PSB) is highly recommended in current agricultural practices that emphasize the adoption of sustainable agriculture. This is because co-inoculation of PSB with symbiotic nitrogen-fixing rhizobia with legumes can produce synergistic effects that optimize the phosphorous and nitrogen availability in the soil, thereby improving plant growth and resilience while reducing dependence on chemical fertilizers [88]. Consequently, several NREs found in the nodules of different legumes have been detected to have the ability to solubilize P and make it available to their host legumes. Examples of such P-solubilizing NREs reported in various legumes include *Pseudomonas* sp. M18 in French bean [89], *Klebsiella pneumoniae* in ground nut [90], *Pseudomonas indica* NSB17 and *Comamonas terrigena* NSB15 in *Glycine max* [74], and *Serratia plymuthica* 33GS in lentils [81]. A detailed list of the diverse non-rhizobial endophytes found in the root nodules of different legumes and the mechanisms by which they promote plant growth is presented in Table 1.

### 6.3. Siderophores

Iron (Fe) is a typical essential plant micronutrient that, in aerobic environments, exists as insoluble hydroxides and oxyhydroxides, which are not accessible to plants and microbes [91]. To overcome this, certain soil microorganisms produce siderophores, low-molecular-weight secondary metabolites, that chelate and sequester iron and make it available to plants [92]. When siderophore-producing rhizobacteria such as the commonly known fluorescent *Pseudomonas* and several other strains colonize the roots, the plants uptake iron (Fe^+3^) from the bacterial siderophores by direct separation from the complex or by exchanging ligands. This uptake of Fe from the bacterial siderophores results in increased plant growth and development [93,94]. In addition to their role in direct plant growth promotion by making Fe available to plants, microbial siderophores also have vital roles in the suppression of phytopathogens, either through competition for iron with the pathogen or by induction of systemic resistance (ISR) [95]. Several strains of non-rhizobial endophytes with the ability to produce siderophores have been isolated from the nodules of various legumes and characterized. For instance, ref. [96] isolated and characterized non-rhizobial endophytes with the ability to produce siderophores from the root nodules of the rooibos tea plant (*Aspalathus linearis* burm f.) in South Africa. The strains mainly belong to *Herbaspirillum seropedicae*, *Herbaspirillum lucitanum*, *Burkholderia pyrocinnia* and *Burkholderia* spp. An in vitro test for the presence of siderophores by these strains resulted in a change in the color of an iron-limited green CAS-agar medium into yellow after sequestering and binding all the available iron (Fe^+3^) (unpublished data) (Figure 5A). These NREs have been isolated from the active nodules of *A. linearis* and are presumably important as nodule endophytes with PGPR properties due to their in vitro siderophore-producing ability (Figure 5B). For further reference, several other examples of siderophore-producing NREs have been listed in Table 1.

### 6.4. Nitrogen Fixation

Several NREs that fix nitrogen in various legume species have so far been reported, although the amount of N fixed is usually lower than that of their rhizobium partners. For example, ref. [97] reported the isolation and characterization of two nodule endophytes from mung bean (*Vigna radiata*) that fixed 19.55 mg N g^−1^ glucose consumed. The isolates, characterized as *Crhysobacterium indologenes* strain LMG 8337 and *C. indologenes* NBRC 1494, also contained a 380 bp fragment of the *nifH* gene, which further validates their nitrogen-fixing ability. The possession of the *nifH* gene in bacteria is considered a good indication of their nitrogen-fixing ability. The *nifH* gene codes for the enzyme nitrogenase reductase, which is a crucial component of the nitrogenase enzyme complex responsible for the fixation of atmospheric nitrogen (N_2_) into ammonia (NH_3_). In a study using the legume *Clitoria ternatea* L. (Asian pigeonwings) by [98], all eight non-rhizobial endophytes that optimally grew on Ashby’s N-free medium over several generations also possessed the *nifH* gene, indicating their capacity to fix nitrogen. Recently, ref. [99] identified and reported other non-rhizobial endophytes, including *Pseudomonas putida* R6, *Pseudomonas proteolytica* GRE6 and *Klebsiella oxytoca* GRE5 from groundnut nodules. In vitro and in vivo tests using these strains indicated that the isolates were able to grow on various N-free media and resulted in an increase in the plant’s nitrogen content.

Among the non-rhizobial endophytes most frequently isolated from the root nodules of legumes, *Bacillus* spp. gain the upper hand. Many strains of *Bacillus* spp. have in the past been reported to be among the most successful co-inoculants when used along with rhizobium inoculants [100,101,102]. In a recent study by [103], *Bacilus aryabatti* strain CM44 and *Bacillus songklensis* strain KCa6 were characterized as endophytic nitrogen-fixing bacteria from the nodules of *Pisum sativum*. Both strains grew optimally on N-free Burk medium and also showed various plant growth-promoting properties, including nitrogen fixation. In addition to *Bacillus* species, several other non-rhizobial endophytes with the capacity to fix atmospheric nitrogen have been characterized from the nodules of *Glycine max*, *Arachis hypogeae*, *Vigna radiata* and *Vigna mungo* [90]. These NREs, which were characterized as *Enterobacter cloacae* AS1, *Chryseobacterium indologenes* AM2, *Enterobacter cloacae*, *Klebsiella pneumoniae* AG4, *Pseudomonas aeruginosa* ABG5, *Enterobacter ludwigii* ABG6 and *Klebsiella variicola* ABG7, possess a ~380 bp fragment of the *nifH* gene. Qualitative estimation of the nitrogen fixation by these strains confirmed the efficiency of the isolates in fixing atmospheric nitrogen in a range of 11.55 to 62.10 mg N g^−1^ of glucose consumed.

## 7. Inoculation Using Consortia of Rhizobia and NREs

One of the strategies to enhance legume nitrogen fixation and crop yield and ensure a sustainable agricultural system involves the co-inoculation of rhizobia with other plant growth-promoting strains of bacteria. Endophytic bacteria, including the non-rhizobial nodule endophytes, live intercellularly within the host tissue and therefore have a more intimate interaction with the host than other plant-associated rhizobacteria [104]. Several legume inoculation trials conducted in the past, under both field and glasshouse conditions, have demonstrated that co-inoculation using rhizobia and other beneficial endophytic bacteria provided better results than single inoculation with rhizobia in terms of the symbiotic effectiveness and crop yield. In a study by [105], co-inoculation of pigeon pea (*Cajanus cajan*) with a mixture of *Rhizobium* sp. and *Bacillus* strains enhanced growth under starved conditions. In another experiment [106], the nodulation and symbiotic effectiveness of *Medicago truncatula* were significantly improved after co-inoculation of the legume with *Ensifer* (*Sinorhizobium*) *medicae* WSM419 and *Pseudomonas fluorescens* WSM3457. The symbiotic effectiveness and growth increase displayed after co-inoculation in both pigeon pea and *Medicago truncatula* described above could be the result of microbial co-operation or synergy between the symbiotic rhizobia and the non-rhizobial strains in the rhizosphere [107]. To elaborate on the microbial co-operation, co-inoculation of *Medicago* sp. with rhizobia and a phosphate-solubilizing endophytic bacteria (PSB) enhanced the nodulation and nitrogen fixation, in parallel with an increase in the plant’s phosphorous (P) content. The improvement in the plant’s P nutrition caused by the PSB was responsible for the increased nitrogen-fixation as these processes are P-dependent [107].

One of the most notable consortia applications in legumes with the potential to improve agricultural sustainability is co-inoculation with a mixture of *Rhizobia* and *Azospirilla* [41]. Apart from being considered an important group of beneficial rhizobacteria in the rhizosphere, *Azospirillum* species have also been detected in the nodules of several legumes [43]. When co-inoculated with rhizobia, these beneficial *Azospirillum* spp. cause changes in the concentration, content and distribution of nutrients in the legumes, as well as changes in the flavonoids, and they protect the legume from various abiotic stresses. In common bean, for instance, co-inoculation of *Rhizobium tropici* and *Azospirillum brasilense* enhanced the drought tolerance as opposed to a single inoculation with the rhizobium [108]. In a more recent co-inoculation study [109], inoculating cowpea with a combination of *Bradyrhizobium* sp. and *Azospirillum brasilense* resulted in an increase in N uptake by cowpea (from 10–14%) and in the grain yield by 8% compared to a single inoculation with *Bradyrhizobium* sp. alone.

Co-inoculation of leguminous plants with rhizobia and other non-rhizobial endophytes is not confined to providing benefits in terms of direct plant growth promotion under normal soil conditions. Many non-rhizobial nodule endophytes carry various beneficial genes and traits that confer tolerance to different abiotic stresses. In a study by [51], the effect of the ACC-deaminase-producing endophytic strain *Pseudomonas* sp. Q1 on the symbiotic effectiveness and growth of the legumes *Trifolium subterraneum* and *Medicago polymorpha* was elucidated after co-inoculation with *Rhizobium leguminosarum* and *Ensifer meliloti* under a high manganese concentration. The result was that, as opposed to a single inoculation with the rhizobia strains, co-inoculation with endophytic *Pseudomonas* sp. Q1 improved the symbiotic performance of *R. leguminosarum* ATCC 14480T and *E. meliloti* ATCC 9930T in subterranean clover and burr medic, respectively, at the same time as contributing to the growth of these pasture legumes under the Mn toxicity condition. A detailed list of the diverse non-rhizobial endophytes found in different legumes and the mechanisms by which they promote growth is presented in Table 1.

**Table 1 microorganisms-13-00518-t001:** Diverse non-rhizobial endophytes found in the root nodules of different legumes and their mechanisms of plant growth promotion.

Species	Phylum	Original Host	Host Tested on	PGP Mechanism	Parameters Promoted	Reference
*Paenibacillus taichungensis*	Bacillota	*Vigna radiata*	*Vigna radiata*	-Production of IAA -Siderophore production -Phosphorus solubilization	-Seedling vigor -Root length -Hypocotyl length -Shoot length -Number of lateral roots -Dry weight	[4]
*Novosphingobium* spp.	Pseudomonadota	*Glycine max*	ND	ND	ND	[10]
*Variovorax* spp.	Pseudomonadota	*Glycine max*	ND	ND	ND	[10]
*Flavobacterium* spp.	Bacteroidota	*Glycine max*	ND	ND	ND	[10]
*Stenotrophomonas* spp.	Pseudomonadota	*Glycine max*	ND	ND	ND	[10]
*Nitrospira* spp.	Pseudomonadota	*Glycine max*	ND	ND	ND	[10]
*Arthrobacter* spp.	Actinomycetota	*Glycine max*	ND	ND	ND	[10]
*Sporosarcina* spp.	Bacillota	*Glycine max*	ND	ND	ND	[10]
*Pseudomonas* spp.	Pseudomonadota	*Pisum sativum*, *Trifolium* sp., *Glycine max*	Peanut, ND, ND	-Heavy metals tolerance	-Root length -Fresh weight	[10]
*Nitrobacter* spp.	Pseudomonadota	*Glycine max*	ND	ND	ND	[10]
*Tardiphaga* spp.	Pseudomonadota	*Glycine max*	ND	ND	ND	[10]
*Bacillus* sp. AAU B6	Bacillota	*Vigna radiata* (*mung bean*)	*Vigna radiata*	-Phosphorus solubilization -Potash mobilization -Production of IAA -1-aminocyclopropane-1-carboxylic acid (ACC) deaminase activity -Nitrogen fixation	-Germination plant height -Number of nodules per plant -Root length -Fresh biomass -Dry biomass -Seed yields	[70]
*Bacillus* sp. AAU B12	Bacillota	*Vigna radiata* (*mung bean*)	*Vigna radiata*	-Phosphorus solubilization -Potash mobilization -Production of IAA -1-aminocyclopropane-1-carboxylic acid (ACC) deaminase activity -Nitrogen fixation	-Germination plant height -Number of nodules per plant -Root length -Fresh biomass -Dry biomass -Seed yields	[70]
*Bacillus* sp. AAU B6	Bacillota	*Vigna radiata* (*mung bean*)	*Vigna radiata*	-Phosphorus solubilization -Potash mobilization -Production of IAA -1-aminocyclopropane-1-carboxylic acid (ACC) deaminase activity -Nitrogen fixation	-Germination plant height -Number of nodules per plant -Root length -Fresh biomass -Dry biomass -Seed yields	[70]
*Comamonas terrigena* NSB15	Pseudomonadota/Proteobacteria	*Glycine max*	*Glycine max*	-Production of IAA -Phosphorous solubilization -Biofilm formation -Cellulase activity -Nitrogen fixation	-Plant dry weight	[74]
*Actinomycetes* spp.	Actinomycetota	*Vicia sativa*, *Glycine max*	*Vicia faba* and *Pisum sativum*, ND	-Production of IAA -Phosphorous solubilization -Protease or cellulase or amylase or chitinase -Antifungal abilities against soil-borne pathogenic fungi	-Shoot fresh weight -Root fresh weight -Root length -Shoot length -Pods -Fresh weight/plant, -Seeds fresh weight/plant -Seeds number	[75]
*Streptomyces variabilis*	Actinomycetota	*Vicia sativa*	*Vicia faba* and *Pisum sativum*	-Production of IAA -Phosphorous solubilization -Protease or cellulase or amylase or chitinase -Antifungal abilities against soil-borne pathogenic fungi	-Shoot fresh weight -Root fresh weight -Root length -Shoot length -Pods weight -Fresh weight/plant -Seeds fresh weight/plant -Seeds number	[75]
*Streptomyces tendae*	Actinomycetota	*Vicia sativa*, *Glycine max*	*Vicia faba* and *Pisum sativum*, ND	-Production of IAA -Phosphorous solubilization -Protease or cellulase or amylase or chitinase -Antifungal abilities against soil-borne pathogenic fungi	-Shoot fresh weight -Root fresh weight -Root length -Shoot length -Pods weight -Fresh weight/plant -Seeds fresh weight/number of seeds -	[75]
*Phyllobacterium ifriqiyense*	Pseudomonadota	*Calobota saharae*, *Calicotome* *villosa*	*Cicer arietinum*	-Production of IAA -Siderophores production	-Shoot dry weight -Root dry weight -Nodules number -Nodules dry weight -Dry weight/nodule -Total nitrogen	[76]
*Xanthomonas translucens*	Pseudomonadota	*Calobota saharae* and *Calicotome* *villosa*	*Cicer arietinum*	-Siderophores production -Production of IAA -Biofilm formation	-Shoot dry weight -Root dry weight -Nodules number -Nodules dry weight -Dry weight/nodule -Total nitrogen	[76]
*Cupriavidus respiraculi*	Pseudomonadota	*Calobota saharae* and *Calicotome* *Villosa*, *Trifolium* sp.	*Cicer arietinum*, ND	-Production of IAA -Siderophores production	-Shoot dry weight -Root dry weight -Nodules number -Nodules dry weight -Dry weight/nodule -Total nitrogen	[76]
*Pantoea dispersa* YBB19B	Pseudomonadota	Groundnut	Groundnut	-Production of IAA, -Siderophore production -Phosphorus solubilization -ACC deaminase activity -Catalase and ascorbate peroxidase activity (antioxidant enzymes)	-Shoot length -Root length -Dry weight of plant -Pod number -Nodule number per plant	[77]
*Bacillus tequilensis* NBB13	Bacillota	Groundnut	Groundnut	-Production of IAA -Siderophore production, -Phosphorus solubilization -ACC deaminase activity -Catalase and ascorbate peroxidase activity (antioxidant enzymes)	-Shoot length -Root length -Dry weight of plant -Pod number -Nodule number per plant	[77]
*Pantoea* spp.	Pseudomonadota	Peanut	Peanut	-Production of IAA, -Siderophore production -Phosphorus solubilization -ACC deaminase activity -Nitrogen-fixation	-Root length -Fresh weights	[78]
*Herbaspirillum* sp.	Pseudomonadota	*Peanut*	Peanut	-Production of IAA -Siderophore production -Phosphorus solubilization -ACC deaminase activity -Nitrogen fixation	-Root length -Fresh weight	[78]
*Blastobacter aggregatus*	Pseudomonadota	*Vigna radiata*	*Vigna radiata*	-Production of IAA -Siderophore production -Phosphorus solubilization	-Seedling vigor -Root length -Hypocotyl length -Shoot length -Number of lateral roots -Dry weight	[80]
*Chitinophaga filiformis*	Bacteroidota	*Vigna radiata*	*Vigna radiata*	-Production of IAA -Siderophore production -Phosphorus solubilization	-Seedling vigor -Root length -Hypocotyl length -Shoot length -Number of lateral roots -Dry weight	[80]
*Dyadobacter fermentans*	Bacteroidota	*Vigna radiata*	*Vigna radiata*	-Production of IAA -Siderophore production -Phosphorus solubilization	-Seedling vigor -Root length -Hypocotyl length -Shoot length -Number of lateral roots -Dry weight	[80]
*Macrophomina phaseolina*	Ascomycota (fungi)	*Vigna radiata*	*Vigna radiata*	-Production of IAA -Siderophore production -Phosphorus solubilization	-Seedling vigor -Root length -Hypocotyl length -Shoot length -Number of lateral roots -Dry weight	[80]
*Blastobacter aggregatus*	Pseudomonadota	*Vigna radiata*	*Vigna radiata*	-Production of IAA -Siderophore production -Phosphorus solubilization	-Seedling vigor -Root length -Hypocotyl length -Shoot length -Number of lateral roots -Dry weight	[80]
*Chitinophaga filiformis*	Bacteroidota	*Vigna radiata*	*Vigna radiata*	-Production of IAA -Siderophore production -Phosphorus solubilization	-Seedling vigor -Root length -Hypocotyl length -Shoot length -Number of lateral roots -Dry weight	[80]
*Dyadobacter fermentans*	Bacteroidota	*Vigna radiata*	*Vigna radiata*	-Production of IAA -Siderophore production -Phosphorus solubilization	-Seedling vigor -Root length -Hypocotyl length -Shoot length -Number of lateral roots -Dry weight	[80]
*Paenibacillus xyla_* *nilyticus*	Bacillota	*Vigna radiata*	*Vigna radiata*	-Production of IAA -Siderophore production -Phosphorus solubilization	-Seedling vigor -Root length -Hypocotyl length -Shoot length -Number of lateral roots -Dry weight	[80]
*Bacillus anthracis*	Bacillota	*Vigna radiata*	*Vigna radiata*	-Production of IAA -Siderophore production -Phosphorus solubilization	-Seedling vigor -Root length -Hypocotyl length -Shoot length -Number of lateral roots -Dry weight	[80]
*Serratia* sp. R6	Pseudomonadota	*Lentils*	Lentils	-Production of IAA -Siderophore production -Phosphorus solubilization -Potassium solubilization -ACC deaminase activity -Nitrogen fixation -Hydrogen cyanide (HCN) -Biofilm production -Protease activity	-Fresh weight -Dry weight -Number of nodules per plant -Nodule fresh weight -Total nitrogen content	[81]
*Bacillus* sp.	Bacillota	*Trifolium* sp.	ND	-Heavy metals tolerance	ND	[110]
*Staphylococcus* sp.	Bacillota	*Trifolium* sp.	ND	-Heavy metals tolerance		[110]
*Enterobacter* sp.	Pseudomonadota	*Trifolium* sp.	ND	-Heavy metals tolerance		[110]
*Acinetobacter* sp.	Pseudomonadota	*Trifolium* sp.	ND	-Heavy metals tolerance	ND	[110]
*Roseomonas* sp.	Pseudomonadota	*Trifolium* sp.	ND	-Heavy metals tolerance	ND	[110]
*Frondihabitans* sp.	Actinomycetota	*Trifolium* sp.	ND	-Heavy metals tolerance	ND	[110]
*Microbacterium* sp.	Actinomycetota	*Trifolium* sp.	ND	-Heavy metals tolerance	ND	[110]
*Kocuria* sp.	Actinomycetota	*Trifolium* sp.	ND	-Heavy metals tolerance	ND	[110]
*Providencia* sp.	Pseudomonadota	*Trifolium* sp.	ND	-Heavy metals tolerance	ND	[110]
*Micrococcus* sp.	Actinomycetota	*Trifolium* sp.	ND	-Heavy metals tolerance	ND	[110]
*Rhodotorula mucilaginosa*	Basidiomycota (fungi)	*Trifolium* sp.	ND	-Heavy metals tolerance	ND	[110]
*Staphylococcus* sp.	Bacillota	*Trifolium* sp.	ND	-Heavy metals tolerance		[110]
*Cupriavidus* sp.	Pseudomonadota	*Trifolium* sp.	ND	-Heavy metals tolerance	ND	[111]
*Bacillus* spp. ESA 417, ESA 418	Bacillota	Cowpea	Cowpea	-Production of auxin -Siderophores -Biofilm formation -Nitrogen fixation	-Grain yield -Root and shoot dry matter -Shoot N content -Nodulation (number and dry matter of nodules)	[111]
*Bacillus* sp. ESA 420	Bacillota	Cowpea	Cowpea	-Production of auxin -Siderophores -Biofilm formation	-Grain yield -Root and shoot dry matter -Shoot N content -Nodulation (number and dry matter of nodules)	[111]
*Chryseobacterium* spp. 29, 23, 19, 412	Bacteroidota	Cowpea	Cowpea	-Production of auxin -Siderophores -Biofilm formation -Nitrogen fixation	-Grain yield -Root and shoot dry matter -Shoot N content -Nodulation (number and dry matter of nodules)	[111]
*Microbacterium* sp. ESA 413	Actinomycetota	Cowpea	Cowpea	-Production of auxin -Siderophores production -Biofilm formation	-Grain yield -Root and shoot dry matter -Shoot N content -Nodulation (number and dry matter of nodules)	[111]
*Agrobacterium* sp. ESA 422	Pseudomonadota	Cowpea	Cowpea	-Production of auxin -Siderophores production -Biofilm formation -Nitrogen fixation	-Grain yield -Root and shoot dry matter -Shoot N content -Nodulation (number and dry matter of nodules)	[111]
*Delftia* ESA 421	Pseudomonadota	Cowpea	Cowpea	-Production of auxin -Siderophores -Biofilm formation	-Grain yield -Root and shoot dry matter -Shoot N content -Nodulation (number and dry matter of nodules)	[111]
*Bacillus* spp.	Bacillota	Cowpea	Cowpea	-Production of auxin -Siderophores -Biofilm formation -Nitrogen fixation	-Grain yield -Root and shoot dry matter -Shoot N content -Nodulation (number and dry matter of nodules)	[111]
*Sphingomonas* sp. ESA 423	Pseudomonadota	Cowpea	Cowpea	-Production of auxin -Siderophores production -Biofilm formation -Nitrogen fixation	-Grain yield -Root and shoot dry matter -Shoot N content -Nodulation (number and dry matter of nodules)	[111]
*Pelomonas* sp. ESA 424	Pseudomonadota	Cowpea	Cowpea	-Production of auxin -Siderophores production -Biofilm formation -Nitrogen fixation	-Grain yield -Root and shoot dry matter -Shoot N content -Nodulation (number and dry matter of nodules)	[111]

ND = not determined.

## 8. Concluding Summary

The root nodule microbiome is mostly occupied by rhizobia and other non-rhizobia strains, most of which have plant growth promotion functions, with symbiotic nitrogen-fixing rhizobia being the predominant population rather than the non-rhizobial endophytes. Rhizobia and other endophytic bacteria residing in the root nodule work synergistically. The culture-independent technique of metagenomics is often used to determine the microbial community (rhizobia and non-rhizobia) in the nodules. Although various NREs, such as *Bacillus*, *Pseudomonas*, *Phyllobacterium* and *Agrobacterium*, are found in the nodules, they enter the nodules at the same time as rhizobia but do not induce nodule formation. The NREs seem to overturn the long-time legume host-specificity dogma that existed for decades and enter the root nodules, with only a limited number of NREs presumably allowed into the interior of the nodules.

Some NREs have the potential to improve legume survival under harsh environmental conditions, while others may be used as bioinoculants for co-inoculation with other rhizobia to enhance rhizobia performance to improve plant growth and yield. Plant growth promotion, environmental stress resistance, disease protection and enhancement of nodule formation are some of the benefits that the host legumes obtain from their endophytic partners. Therefore, NREs promote plant health and plant growth by eliminating diseases, making nutrients accessible to plants, and preventing plant abiotic stresses. Co-inoculation of legumes with rhizobia and NREs yields positive results in most studies, including significant increases in the rate of root nodule development and plant biomass and improved symbiotic performance. The tripartite association yields beneficial solutions to improve legume–rhizobium symbiosis. The idea of developing consortia of *Rhizobium* and other beneficial non-rhizobial endophytic strains in a single biofertilizer product could be a very useful approach to promote plant growth under abiotic stress conditions, particularly drought, salt, acidity, waterlogging and other stresses associated with metal toxicity. As there are different views and assumptions concerning how exactly non-rhizobial endophytes make their way into the interior of the legume nodules from the rhizosphere of the root zone, this field of study warrants more comprehensive research accompanied by high-throughput cellular, microscopic and molecular techniques.

## Figures and Tables

**Figure 1 microorganisms-13-00518-f001:**
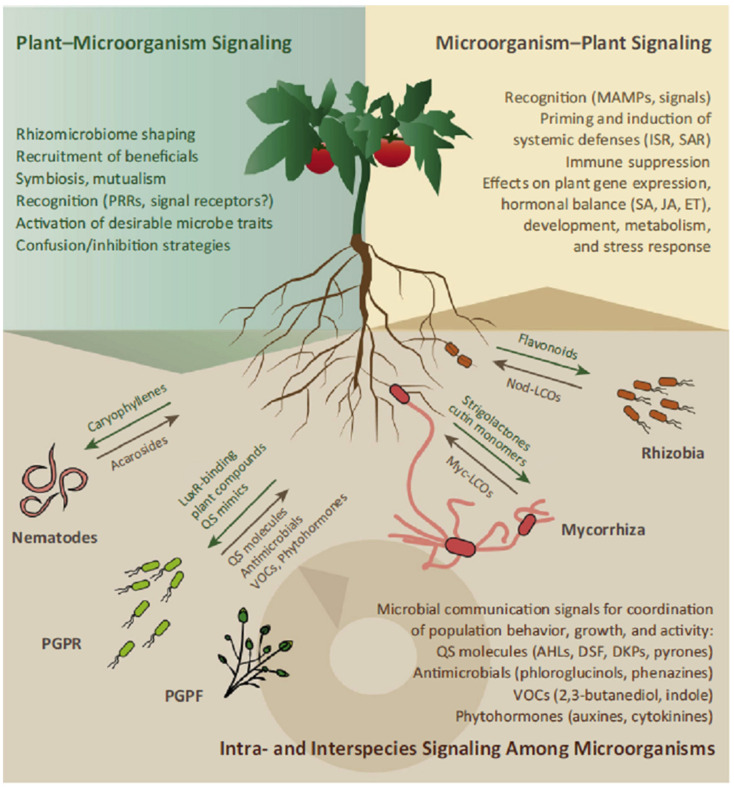
Signaling among microorganisms and inter-kingdom signaling between microorganisms and plants in the rhizosphere. Source [18] with permission.

**Figure 2 microorganisms-13-00518-f002:**
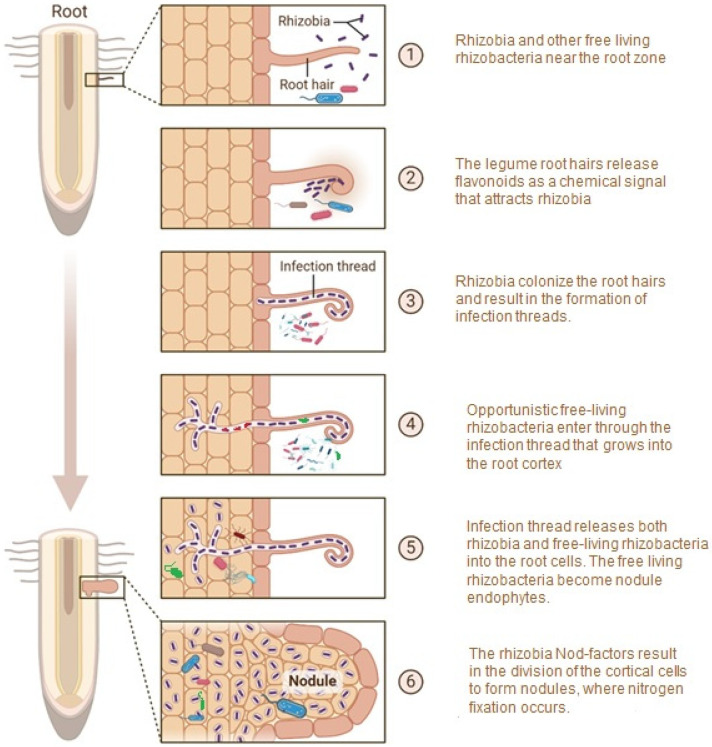
One of the proposed mechanisms of how non-rhizobial endophytes enter the root nodules and become part of the nodule microbiome that includes both the symbiotic rhizobia and the once free-living rhizobacteria, which were part of the rhizosphere soil. Rhizobium species are colored in purple, while free-living NRE are colored in red, blue and green. Created with BioRender.com [32].

**Figure 3 microorganisms-13-00518-f003:**
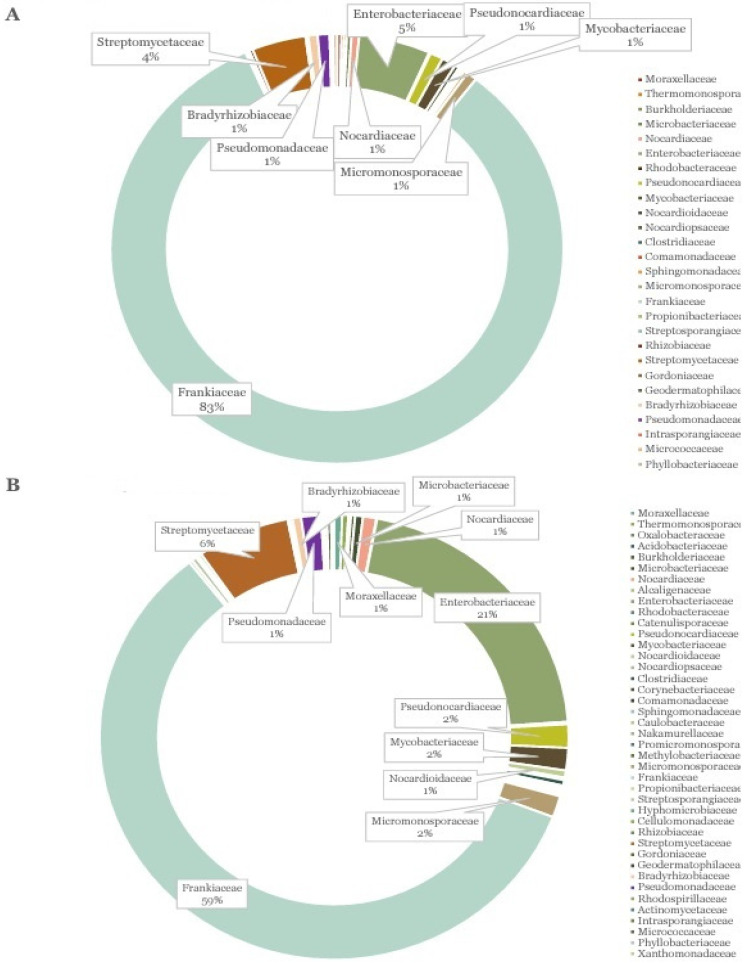
Microbial diversity of the nodule microbiome of Actinorhizal plants determined using shotgun metagenomics. The core microbiome of the nodules retrieved from greenhouse (**A**) and field (**B**) samples, comprising a total of 27 and 41 families, respectively. Source: [37] with permission.

**Figure 4 microorganisms-13-00518-f004:**
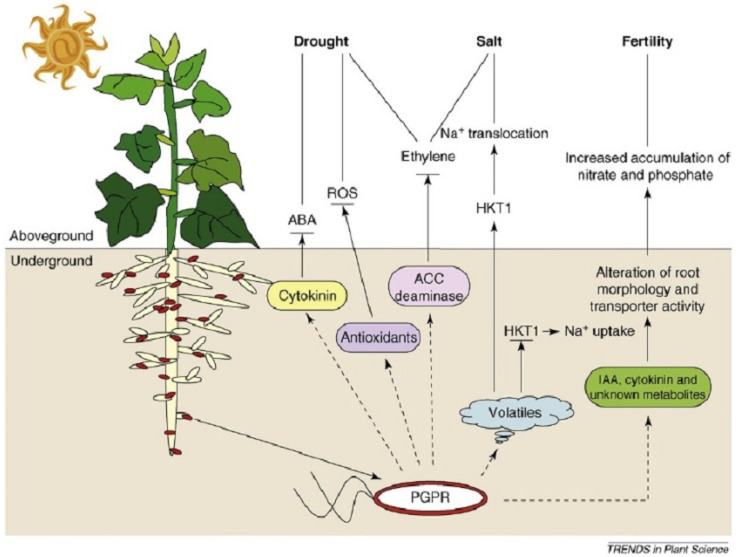
A schematic representation of how PGPR are involved in the mitigation of various abiotic stresses, including drought, salinity, and nutrient deficiency (fertility) stress, and elicit induced systemic tolerance (IST) in plants. Source: [65] with permission.

**Figure 5 microorganisms-13-00518-f005:**
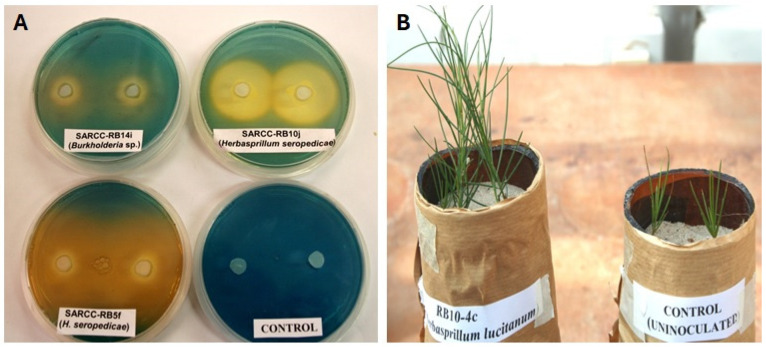
(**A**) In vitro siderophore production by non-rhizobial endophytes *Herbaspirillum seropedicae* and *Burkholderia* sp. isolated from the nodules of *Aspalathus linearis* (rooibos). (**B**) *Aspalathus linearis* inoculated with the non-rhizobial endophytic strain *Herbaspirillum lucitanum* (**left**) as compared to the non-inoculated control (**right**). Source: [96].

## Data Availability

The original contributions presented in the study are included in the article, further inquiries can be directed to the corresponding author.

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
