# Peer review of "Non-Rhizobial Endophytes (NREs) of the Nodule Microbiome Have Synergistic Roles in Beneficial Tripartite Plant–Microbe Interactions"

_microorganisms, 2025, doi:10.3390/microorganisms13030518_

Round 1

Reviewer 1 Report

Comments and Suggestions for Authors

Review on “Non-rhizobial endophytes (NREs) of the nodule microbiome have synergistic roles in beneficial tripartite plant-microbe interactions” for manuscript ID microorganisms-3333405

In brief introduction the authors emphasize the significance of understanding microbial communities in agriculture and their potential benefits for sustainable practices. The aim of the following review is to evaluate the role of NREs in symbiosis with legumes and study abundance of NREs’ in root nodules.

This review has several major flaws:

Literature review methodology isn’t described;

The scope of species (among legumes) included in review is superficial;

Some parts of the text are taken from authors’ the previous book;

Concluding section should highlight the knowledge gap or show the future research direction, but avoid just re-telling the facts (L510, L523)

There are plenty of studies covering NREs and their relationships with legumes, moreover, the recent review close to the topic is available: Hnini M and Aurag J (2024) Prevalence, diversity and applications potential of nodules endophytic bacteria: a systematic review. Front. Microbiol. 15:1386742. http://doi.org/10.3389/fmicb.2024.1386742

By the way, some recent studies have been missed. The following papers could help to improve the paper:

·        Bakhtiyarifar, M., Enayatizamir, N. & Mehdi Khanlou, K. Biochemical and molecular investigation of non-rhizobial endophytic bacteria as potential biofertilisers. Arch Microbiol 203, 513–521 (2021). https://doi.org/10.1007/s00203-020-02038-z

·        Muindi, M. M., Muthini, M., Njeru, E. M., & Maingi, J. (2021). Symbiotic efficiency and genetic characterization of rhizobia and non rhizobial endophytes associated with cowpea grown in semi-arid tropics of Kenya. Heliyon, 7(4). https://doi.org/10.1016/j.heliyon.2021.e06867

·        Martínez-Hidalgo, P., Humm, E.A., Still, D.W. et al. Medicago root nodule microbiomes: insights into a complex ecosystem with potential candidates for plant growth promotion. Plant Soil 471, 507–526 (2022). https://doi.org/10.1007/s11104-021-05247-7

·        Sameh H Youseif, Fayrouz H Abd El-Megeed, Ali S Abdelaal, Amr Ageez, Esperanza Martínez-Romero, Plant–microbe–microbe interactions influence the faba bean nodule colonization by diverse endophytic bacteria, FEMS Microbiology Ecology, https://doi.org/10.1093/femsec/fiab138

Major points:

Section 5, L223-311, Section 6: L450-475: part of the text taken from authors’ the previous book https://doi.org/10.1007/978-81-322-2647-5_2

Intro part to metagenomics could be omitted in Section 4 (L190-L201).

Minor points:

L41, L36: reference is required

L56: “Lipochitooligosaccharides” → “lipo-chitooligosaccharides”

Bulk citations aren’t good practice (L346, L363), please select the most suitable references supporting your point.

L342: “)” to “]”

L519: “novel innovation” – please rephrase the sentence

Author Response

Response to the reviewers

Reviewer 1

Dear reviewer, many thanks for the comments and suggestions on our submitted review manuscript. We greatly value your esteemed comments, and we believe they have helped us to greatly improve the content and the overall scientific looks to attract the attention of reader researchers and scientists in the field. We thank you very much. Below are point by point responses to your comment.

  1. Literature review methodology is not described

Response: We have included a brief methodology of literature review.

  1. The scope of species (among legumes) included is superficial.

Response: We have included as much scope as we could by including additional legume species and their microsymbionts.

  1. The conclusion section should highlight the knowledge gap or show the future research direction,  but just avoid telling the facts.

Response: We have made improvement in the conclusion as per the suggestion.

  1. Suggested literature reviews by reviewer.

Response: We have included all the suggested additional literature reviews and improved the content of our manuscript.

Major points

  1. Section 5, L233-311, SECTION 6: l450-475; part of the text taken from author’s publications.

Response: We have omitted (deleted) all parts taken from our previous work and have modified the contents by adding new information.

  1. Intro part of metagenomics could be omitted.

Response: We have deleted the introductory part of metagenomics.

Minor Points

  1. L41, L36. Reference is required

Response: We have added references (1 &2)

  1. L56, Lipochitooligosaccharides should be changed to ‘ lipo-chitooligisaccharides’

Response. Changed as suggested

  1. Bulk citations are not good practice.

Response. We have minimized and selected the most suitable ones.

  1. Typo error

Response: Correction made

  1. L519, Novel innovation, please rephrase the sentence

Response: rephrased as suggested.

Reviewer 2 Report

Comments and Suggestions for Authors

This is an interesting and up-to-date review adding additional data on the occurrence and potential function of non-Rhizobial endophytes in the host-Rhizobium symbiosis. I have some minor comments on phrasing to improve readability. In Table 1, some references do not cover the information on IAA synthesis (reference number color marked in pdf), whereas for some papers IAA and others, auxin is used. All papers cited use the Salkowski colorimetric assay for auxin detection, that is the reason to maybe choose for the term auxin production instead of IAA production. The most important fact is that it is the same term used in all of them, not to confuse the reader. The Latin name of the host should also be in italic.

reference 37, Umea is in Sweden, not in Germany

reference 81: this reference is incomplete, journal is missing

all comments marked in the pfd attached

Author Response

Dear reviewer,

Many thanks for your valuable comments. The team has worked through the comments, and we have made all the suggested revisions.

Commet1. In Table 1, some references do not cover the information on IAA synthesis (reference number color marked in pdf), whereas for some papers IAA and others, auxin is used. All papers cited use the Salkowski colorimetric assay for auxin detection, that is the reason to maybe choose for the term auxin production instead of IAA production.

Response. This is a valid comment that we value a lot. However, since we believe the term auxin is more frequently referred for the plant phytohormone, we choose to use the term IAA for the bacterial hormone as it appears in several microbiology publications. And as suggested, we made it consistent throughout the manuscript text.

Comment 2:  The Latin name of the host should also be in italic.

Response: We have checked all host names and changed into italics.

Comment 3; reference 37, Umea is in Sweden, not in Germany

Response: This reference is omitted due to other reviewer's comment to remove some of the contents in the manuscript

Comment 4: reference 81: this reference is incomplete, journal is missing

Response: Since this is a preprint on Research Square, it is omitted 

all comments marked in the pfd attached

End of response

Reviewer 3 Report

Comments and Suggestions for Authors

Generally well written review of an interesting addition to the rhizobial literature. Only a few minor editing changes.

Line 122 - change to "...soil phosphorus, whereas other..."

Line 245 - delete 'The stress'

Line 273 - change 'so-called global warming' to 'climate change'

Line 340, 401, Table 1 - change to Glycine max

Author Response

Dear Reviewer, 

Thank you so much for your comments. We value your comments and have made all necessary changes as suggested.

Line 122 - change to "...soil phosphorus, whereas other..."  

Response: This section was omitted, so no correction to make 

Line 245 - delete 'The stress'

Response. Deleted

Line 273 - change 'so-called global warming' to 'climate change'

Response: changed

Line 340, 401, Table 1 - change to Glycine max

Response: Changed

End of response

Round 2

Reviewer 1 Report

Comments and Suggestions for Authors

At first, I would like to thank the authors for the improving the manuscript, but some concerns remain to be addressed.

Major points:

Unfortunately, all figures are missing in the submitted PDF file, so the previous manuscript version was used.

By the way, a lot of new literature sources were added to the manuscript, but Table 1 wasn't extended or ever changed.

Minor points:

References order is not continuous

L99: PLOS is not a search engine, but company publishing several journals. 

L110: no need to capitalize letters and italic font here, add abbreviation (QS)

L147: "non-rhizobial bacteria were isolated": reference is required

L181: "researchers indicated": please rephrase

L187: gene names taken incorrectly from [8], please re-check

L190: remove the dot and space

L195-196, 211, 216: obsolete or incorrect taxons names

L233: capital letters are unnecessary here

Author Response

Response to reviewers

Dear reviewer/s, many thanks for your valuable comments and reviews on our submitted manuscript. We greatly value your comments and below, we provide a point-by-point response to the comments:

Minor Points.

  1. L-99. PLOs is not a search engine.

Response:

We have deleted it from the text

  1. L-110, no need to capitalize letters and italic font here. Add abbreviations (QS)

Response:

We accept the comment and have corrected accordingly.

  1. L-147- Non rhizobial bacteria were isolated. Reference is required here.

Response:

Reference is provided for this.

  1. L-181. Researchers indicated. Please rephrase.

Response:

We have rephrased it.

  1. L-187. Gene names taken incorrectly from 8.

Response:

We have rectified the mistakes and replaced the correct ones.

  1. L-190. Remove the dot and space.

Response:

Corrected accordingly

  1. L-195-196. Obsolete, incorrect taxon names.

Response:

We agree with the comment and have changed into more acceptable taxon names

  1. L-233. Capital letters are unnecessary here.

Response.

We have made the correction

Major Points

  1. Unfortunately, all figures are missing in the submitted PDF file.

Response:

We are sorry for that. However, we submitted all the figures in separate files and the main manuscript without figures in another file. We have now rectified that error, and we have included all the figures in the main manuscript, as well as separate files.

  1. Reference order is not continuous

Response:

We agree with this comment, and we have worked on the reference list again so that it becomes continuous. This was the part that we struggled with a lot, since in trying to make it continuous, the entire manuscript text gets disorganized. Therefore, we took time to make this feasible and have made all the necessary steps and corrections. There might be a few references, which are still not in continuous manner, due to the Table citation, or important (same) references cited more than once  for different aspects of studies.

  1. BTW, new literature sources were added to the manuscript, but Table 1 is not extended or ever changed.

Response:

We value this comment, and we have made all the required changes to be made. On this occasion, we would like to inform the reviewers again that , even in this correction we have added new references, in order to cope with the continuity of the reference citations, and some old ones removed.

Response to reviewers

Dear reviewer/s, many thanks for your valuable comments and reviews on our submitted manuscript. We greatly value your comments and below, we provide a point-by-point response to the comments:

Minor Points.

  1. L-99. PLOs is not a search engine.

Response:

We have deleted it from the text

  1. L-110, no need to capitalize letters and italic font here. Add abbreviations (QS)

Response:

We accept the comment and have corrected accordingly.

  1. L-147- Non rhizobial bacteria were isolated. Reference is required here.

Response:

Reference is provided for this.

  1. L-181. Researchers indicated. Please rephrase.

Response:

We have rephrased it.

  1. L-187. Gene names taken incorrectly from 8.

Response:

We have rectified the mistakes and replaced the correct ones.

  1. L-190. Remove the dot and space.

Response:

Corrected accordingly

  1. L-195-196. Obsolate, incorrect taxon names.

Response:

We agree with the comment and have changed into more acceptable taxon names

  1. L-233. Capital letters are unnecessary here.

Response.

We have made the correction

Major Points

  1. Unfortunately, all figures are missing in the submitted PDF file.

Response:

We are sorry for that. However, we submitted all the figures in separate files and the main manuscript without figures in another file. We have now rectified that error, and we have included all the figures in the main manuscript, as well as separate files.

  1. Reference order is not continuous

Response:

We agree with this comment, and we have worked on the reference list again so that it becomes continuous. This was the part that we struggled with a lot, since in trying to make it continuous, the entire manuscript text gets disorganized. Therefore, we took time to make this feasible and have made all the necessary steps and corrections. There might be a few references, which are still not in continuous manner, due to the Table citation, or important (same) references cited more than once  for different aspects of studies.

  1. BTW, new literature sources were added to the manuscript, but Table 1 is not extended or ever changed.

Response:

We value this comment, and we have made all the required changes to be made. On this occasion, we would like to inform the reviewers again that , even in this correction we have added new references, in order to cope with the continuity of the reference citations, and some old ones removed.
